# Using Wastewater Surveillance to Monitor Gastrointestinal Pathogen Infections in the State of Oklahoma

**DOI:** 10.3390/microorganisms11092193

**Published:** 2023-08-30

**Authors:** Katrin Gaardbo Kuhn, Rishabh Shukla, Mike Mannell, Grant M. Graves, A. Caitlin Miller, Jason Vogel, Kimberly Malloy, Gargi Deshpande, Gabriel Florea, Kristen Shelton, Erin Jeffries, Kara B. De León, Bradley Stevenson

**Affiliations:** 1Department of Biostatistics & Epidemiology, University of Oklahoma Health Sciences Center, Oklahoma City, OK 73104, USA; kimberly-malloy@ouhsc.edu (K.M.); gargi-deshpande@ouhsc.edu (G.D.); 2School of Civil Engineering and Environmental Science, University of Oklahoma, Norman, OK 73019, USA; rishabh.shukla@ou.edu (R.S.); ggraves@usgs.gov (G.M.G.); caitlin.miller@ou.edu (A.C.M.); jason.vogel@ou.edu (J.V.); gabriel-florea@omrf.org (G.F.); 3Acute Diseases Division, Oklahoma State Department of Health, Oklahoma City, OK 73102, USA; mikem@health.ok.gov; 4Oklahoma Medical Research Foundation, Oklahoma City, OK 73104, USA; 5School of Biological Sciences, University of Oklahoma, Norman, OK 73019, USA; kristen.k.shelton-1@ou.edu (K.S.); erin.jeffries@ou.edu (E.J.); deleonkb@ou.edu (K.B.D.L.); 6Department of Earth and Planetary Sciences, Northwestern University, Evanston, IL 60208, USA; bradley.stevenson@ou.edu

**Keywords:** wastewater surveillance, gastrointestinal pathogens, foodborne infections, outbreaks, seasonality, monitoring, surveillance

## Abstract

During the COVID-19 pandemic, wastewater surveillance was widely used to monitor temporal and geographical infection trends. Using this as a foundation, a statewide program for routine wastewater monitoring of gastrointestinal pathogens was established in Oklahoma. The results from 18 months of surveillance showed that wastewater concentrations of *Salmonella*, *Campylobacter,* and norovirus exhibit similar seasonal patterns to those observed in reported human cases (F = 4–29, *p* < 0.05) and that wastewater can serve as an early warning tool for increases in cases, offering between one- and two-weeks lead time. Approximately one third of outbreak alerts in wastewater correlated in time with confirmed outbreaks of *Salmonella* or *Campylobacter* and our results further indicated that several outbreaks are likely to go undetected through the traditional surveillance approach currently in place. Better understanding of the true distribution and burden of gastrointestinal infections ultimately facilitates better disease prevention and control and reduces the overall socioeconomic and healthcare related impact of these pathogens. In this respect, wastewater represents a unique opportunity for monitoring infections in real-time, without the need for individual human testing. With increasing demands for sustainable and low-cost disease surveillance, the usefulness of wastewater as a long-term method for tracking infectious disease transmission is likely to become even more pronounced.

## 1. Introduction

The classic passive surveillance approach for infectious diseases relies on patients developing symptoms, seeking medical care, undergoing testing, diagnosis, and ultimately their case being reported to authorities (for a reportable disease) and suffers from drawbacks with timeliness and accuracy [1]. Most cases are not reported until several weeks after infection and symptom onset and many others are missed because of a lack of medical attention or testing. In addition, a certain proportion of infections are asymptomatic and will never be diagnosed and reported unless accidentally caught during unrelated testing procedures. As such, there is strong consensus that routinely collected infectious disease surveillance data only represent a small fraction of actual infections occurring in the community [1,2].

Although wastewater surveillance of infectious diseases has been used for several decades, it reached an unprecedented scale of use during the COVID-19 pandemic, either as a complement to or in some cases a replacement for existing surveillance strategies [3,4,5]. Until the pandemic, wastewater surveillance was often used as a research tool for specific pathogens or biomarkers; however, there were few published examples of routine surveillance of infectious disease trends using wastewater. These include a program to monitor the introduction of wild poliovirus in Israel [6]; pilot studies for enteric viruses and waterborne pathogens in France, California, and Costa Rica; and isolated studies to detect various pathogens over a limited shorter time period [7,8,9,10,11,12]. The lessons learned from the COVID-19 pandemic in relation to wastewater have been numerous and overall suggest that there is a strong potential for this surveillance to be employed as part of a standard monitoring procedure for infectious diseases to inform public health action [3]. This is primarily based on the fact that wastewater surveillance offers several advantages because it captures the infections as they occur in real-time (as soon as pathogens are excreted in an infected person’s feces or urine) and because it also includes those people who are infected but not diagnosed and reported through the traditional surveillance approach.

The foodborne pathogens *Salmonella* spp., *Campylobacter* spp., and norovirus are among the most reported causes of gastrointestinal illness in humans. Diagnosed infections of *Salmonella* and *Campylobacter* are mandatorily notifiable to state health authorities in the US. They usually follow a seasonal pattern where sporadic cases and outbreaks are reported throughout the year, with peak activity occurring during the summer months and holidays such as Thanksgiving and Christmas [13,14,15]. While norovirus is not a notifiable disease, it is the most frequently reported cause of gastrointestinal disease outbreaks in settings where large numbers of people gather, including cruise ships, schools, care homes, and restaurants. Similar to the bacterial foodborne pathogens, norovirus also exhibits a seasonal pattern where infections peak during late fall to early spring [16]. Although robust surveillance systems exist for *Campylobacter* and *Salmonella*, and norovirus outbreaks are often extensively investigated using human testing, it is widely recognized that knowledge about the transmission and true burden of these pathogens is limited by the current passive surveillance approach. Because wastewater surveillance is entirely independent of human testing and captures all persons in a community, regardless of their symptomatic stage, it is an excellent candidate for actively monitoring the true extent and timing of foodborne pathogen transmission. For example, wastewater has previously been used to detect short-term outbreaks of *Salmonella* and *Campylobacter* outbreaks in military camps in Norway [17], investigate the presence of norovirus in a single city in Sweden [18], and correlate wastewater findings with human cases in a Salmonella outbreak in Texas [19]. However, there is little published evidence to show that wastewater-based surveillance has been used for long-term monitoring of foodborne pathogen transmission with the aim of routinely supplementing traditional surveillance approaches. Given the estimated extensive burden of these diseases, a timely and more population representative monitoring program has tremendous potential for better understanding the transmission dynamics of foodborne infections and for direct public health applications.

The University of Oklahoma Wastewater Surveillance program was established in 2020 as a response to the COVID-19 pandemic [5]. Initially the program covered SARS-CoV-2 surveillance in dorm buildings at the University of Oklahoma, Norman campus; however, the number of locations and pathogens expanded through 2021 and 2022. At the end of 2022, the program included monitoring locations across the state of Oklahoma and routine testing for a range of gastrointestinal and other respiratory viruses. In this paper we describe the results from routine wastewater surveillance of three different gastrointestinal pathogens in the state of Oklahoma, with the primary goals of assessing the potential for long-term monitoring of these pathogens in human populations through wastewater and reporting findings from the surveillance in comparison to known or suspected outbreaks of gastrointestinal illness.

## 2. Methods

### 2.1. Wastewater Sampling

Wastewater treatment plant influent samples were collected routinely between June 2021 and December 2022, mainly as time-weighted composite samples, as described in detail by Kuhn et al. [5]. The number of locations included in the surveillance program increased from 10 wastewater treatment plants in 4 cities (Oklahoma City, Tulsa, Norman and Anadarko) in July 2021 to 32 plants that treat wastewater for approximately 1.8 million persons in 27 cities at the end of December 2022. In each location, the primary wastewater treatment plant serving the highest proportion of the city population was selected for monitoring (if the city held more than one treatment plant).

The samples were collected twice per week for Oklahoma City and Tulsa sites between November 2021 and August 2022 and once per week for all other sites and times. All samples utilized proper chain-of-custody forms. Tubing, connectors, autosampler bottles, and strainers were sanitized in a working solution of 50 mg/L (0.005%) sodium hypochlorite, rinsed with sterile sodium thiosulphate solution (prepared as 1 mL of 10% stock per liter of water) after each sample collection to remove remaining chlorine and ensure no contamination of samples, and finally thoroughly rinsed with reverse osmosis water [20]. All samples were kept between 1 and 6 °C and processed within 24 h of collection.

### 2.2. Genetic Material Extraction and Quantification

Detection and quantification of bacterial and viral particles in the wastewater samples were performed on triplicate 32 mL subsamples that were strained through a 70 μm nylon mesh cell strainer into centrifuge tubes containing 8 mL of a 5× PEG:NaCl solution (62.5 mM PEG8000, 1 M NaCl) as described by Kuhn et al. [5]. We added 100 µL of a 1000-fold dilution of a vaccine containing Bovine Coronavirus (BCoV, Zoetis CALF-GUARD^®^ Bovine Rota-Coronavirus Vaccine) prior to sample extraction to assess the efficiency of the sample processing method. After spiking with BCoV, the samples were vortexed for 15–30 s, incubated overnight at 4 °C, and then concentrated by centrifugation at 14,600× *g* for 45 min at 4 °C. The supernatant was decanted and the pelleted solids were used for total nucleic acid extraction following a protocol modified from the Bio On Magnetic Beads platform [21], described in Kuhn et al. [5].

Quantitative PCR (qPCR) assays were used to detect *Campylobacter* and *Salmonella* DNA in the samples and a Reverse Transcriptase Quantitative PCR (RT-qPCR) assay was used to detect norovirus genotype group II RNA genomes, utilizing primers and TaqMan probes specific for genes or genome regions of the targets [22,23,24] (Appendix A). All reactions were run in triplicate and contained 1× TaqPath™ 1-Step RT-qPCR Master Mix (Thermo Scientific, Waltham, MA, USA), forward and reverse primers, TaqMan probe, and 5 µL of template (1:4 dilution of extracted nucleic acids) in a final volume of 25 µL. The quantity of *Campylobacter* and *Salmonella* targets was estimated using a standard curve containing 5 µL of 10^1^, 10^2^, 10^3^, 10^4^, and 10^5^ copies of a DNA fragment (gBlock, IDT) in triplicate. Norovirus quantity was estimated using a standard curve containing 5 µL of 10^2^, 10^3^, 10^4^, 10^5^, and 10^6^ copies of a synthetic DNA fragment (gBlock, IDT) in triplicate. The sequences for primers, probes, and synthetic DNA controls, along with thermocycling conditions, are described in Appendix A.

### 2.3. Human Cases and Outbreaks

The weekly number of reported cases of *Campylobacter* spp. and *Salmonella* spp. from July 2021 to December 2022 was obtained from the Oklahoma State Department of Health (OSDH) through the statewide notification system for reportable infectious diseases, the Public Health Investigation and Disease Detection of Oklahoma System (PHIDDO). We obtained information on local confirmed outbreaks of *Campylobacter* spp. and *Salmonella* spp. as well as gastrointestinal outbreaks of unknown etiology from the OSDH which are routinely reported to the epidemiology team from institutional settings or foodborne complaints. Information on multistate outbreaks of *Campylobacter* spp. and *Salmonella* spp. was acquired from the Centers for Disease Control and Prevention (CDC) [25]. Because norovirus is not a state-reportable condition in Oklahoma routine surveillance data are not available. The timing of confirmed outbreaks of norovirus was instead collected through information provided by the CDC [25].

### 2.4. Statistical Methods

We calculated the statewide weekly average wastewater concentration of norovirus, *Campylobacter,* and *Salmonella* spp. (viral or bacterial copies per liter of wastewater), including three-week moving averages, for the surveillance period of July 2021 through December 2022. We analyzed seasonal trends in wastewater concentrations and reported cases using basic regression with time series data (week and quarter). Comparisons of trends in reported cases and wastewater concentrations were undertaken using a bivariate time series analysis in a cross-correlation matrix, adjusted for the impact of seasonality, with wastewater concentrations as the dependent and reported cases as the independent variable. For each pathogen, we calculated within season ‘outbreak thresholds’ defined as the 95th percentile of data points within each season as described by the World Health Organization [26,27]. For this purpose, seasons were defined as winter (December–February), spring (March–May), summer (June–August), and fall (September–November). We compared wastewater outbreak signals temporally to notifications of confirmed outbreaks of either pathogen as well as outbreaks of unknown etiology as described above.

All statistical analyses were performed in STATA version 17.0 [28].

## 3. Results

From 13 July 2021 to 31 December 2022, we monitored wastewater across the state of Oklahoma for the presence of *Salmonella* spp., *Campylobacter* spp., and norovirus GII. Due to scheduled breaks in the routine surveillance, concentrations of *Campylobacter* and *Salmonella* were not available between week 42 (10 October) and week 50 (5 December) in 2021 and for norovirus between week 29 (11 July) and week 39 (19 September) in 2021.

### 3.1. Wastewater Concentrations and Cases over Time

During the study period, a total of 2616 cases of foodborne bacterial infections were reported in the state of Oklahoma with the majority of infections (56%) being *Campylobacter* (Table 1). The reported cases and wastewater concentrations of *Campylobacter* and *Salmonella* and wastewater concentrations of norovirus (no reported cases) fluctuated over time with peaks at certain points throughout the year (Figure 1, Figure 2 and Figure 3, Table 1). Cases and wastewater concentrations of *Salmonella* and *Campylobacter* exhibited a significant seasonal pattern (F = 4–29, *p* < 0.05) with highest activity during the summer months and brief peaks in late fall and winter. Although concentrations of norovirus were, on average, higher during late fall to early spring (Figure 3), there was no significant seasonal pattern in the data (F = 1, *p* = 0.26). The cross-correlation time series analysis showed moderate correlations between wastewater concentrations and reported cases (correlation coefficient 0.51–0.58). For *Campylobacter*, the peak correlation was observed at minus 2 weeks indicating that concentrations of *Campylobacter* in wastewater were positively correlated with reported *Campylobacter* cases two weeks later. For *Salmonella*, wastewater concentrations were positively correlated with reported cases one week later.

### 3.2. Outbreaks

During the study period, there were 12 outbreaks of *Salmonella* and *Campylobacter* reported in the information sources used (the first ‘outbreak indicator’ highlighted as shaded boxes in Figure 1, Figure 2 and Figure 3). Of these, two were local to Oklahoma and three were multistate outbreaks with reported cases in Oklahoma. For the five outbreaks with cases in Oklahoma, four were *Salmonella* spp. and one was *Campylobacter* spp. These five outbreaks cumulatively lasted 53 weeks (Figure 1, Figure 2 and Figure 3, Table 2). No laboratory-confirmed norovirus outbreaks occurred during the study period whereas a total of 15 gastrointestinal outbreaks lasting 22 weeks were reported in Oklahoma without the causative pathogen being identified (timing not shown).

Using the defined seasonal thresholds of 95th percentiles, we observed 28 weeks where wastewater concentrations indicated an ‘outbreak’—i.e., exceeding the seasonal 95th percentile (Table 2). The seasonal thresholds represent the second ‘outbreak indicator’ in Figure 1, Figure 2 and Figure 3, shown as dotted lines in each season for each pathogen. Of the twenty-eight ‘outbreak’ weeks, nine (32%) matched in time with confirmed *Salmonella* or *Campylobacter* outbreaks reported no later than two weeks after the exceeded wastewater threshold. Out of eleven weeks with *Salmonella* ‘wastewater outbreak’ alerts, nine (82%) matched in time with a reported outbreak. One wastewater alert for *Salmonella* matched in time with an outbreak of unknown etiology (Table 2). None of the wastewater alerts for *Campylobacter* matched in time with a confirmed campylobacteriosis outbreak; however, two weeks of elevated *Campylobacter* concentrations in wastewater correlated with a reported gastrointestinal outbreak of unknown etiology (Table 2). Of the twenty-two weeks with reported gastrointestinal outbreaks without a confirmed pathogen, five (23%) matched in time with an outbreak alert for norovirus alone (Table 2). Overall, 41% of the weeks with a confirmed gastrointestinal outbreak of unknown etiology matched in time with wastewater alerts for either *Salmonella*, *Campylobacter,* or norovirus (Table 2).

Of the multistate outbreaks with no reported cases in Oklahoma, we found that two matched in time with exceeded wastewater thresholds for the outbreak pathogen (Table 3). Specifically, wastewater thresholds indicated one *Salmonella* and one norovirus outbreak which matched in time with reported multistate outbreaks of the same pathogens (Figure 1, Figure 2 and Figure 3, Table 2).

Finally, there were a number of wastewater outbreak alerts which did not match in time with reported gastrointestinal outbreaks in either Oklahoma or multistate occurrences. For *Campylobacter*, this was a total of eight weeks (80%) of wastewater alerts, while it was one week (9%) for *Salmonella* and two (29%) for norovirus.

## 4. Discussion

In this paper we describe the results of an 18-month program for wastewater surveillance of *Salmonella* spp., *Campylobacter* spp., and norovirus pathogens in the state of Oklahoma. Numerous studies have demonstrated the utilization of wastewater for monitoring infectious pathogens however, the vast majority have focused on SARS-CoV-2. To our knowledge, this is one of the first studies which demonstrates the feasibility of routine monitoring for gastrointestinal pathogens in wastewater over a large geographical scale and establishes temporal correlations between pathogen concentrations in wastewater and reported human cases and outbreaks during a longer time period By comparing trends in wastewater concentrations to reported human cases and confirmed outbreaks of the same pathogens, we present evidence that wastewater monitoring is a promising real-time alternative, or supplement, to traditional infectious disease surveillance based on passive case reporting.

Both reported cases and wastewater concentrations of *Campylobacter* spp. and non-typhoidal *Salmonella* spp. in Oklahoma exhibited significant seasonal patterns, comparable to those generally reported from traditional surveillance [13,14,29]. Human infections of norovirus (although not mandatorily notifiable) traditionally peak in late fall to early spring and, while we did not detect a significant seasonal pattern in norovirus wastewater concentrations, the highest concentrations in general were observed from January until May. For all three pathogens, the tendency of wastewater concentrations to follow known seasonal patterns of human illnesses is an encouraging first sign that wastewater surveillance can effectively be applied to monitor trends in gastrointestinal infections. Our results demonstrating seasonality of gastrointestinal pathogens in wastewater correlate with those previously reported for norovirus and *Salmonella* spp. [7,10,30] with the added advantage of spanning several seasons and multiple locations representing diverse population groups. Using time series analysis, we showed that wastewater concentrations were significantly correlated with reported cases of *Salmonella* and *Campylobacter* one and two weeks later, respectively. Generally, this suggests that, for these two pathogens, wastewater concentrations can serve as an early warning indicator of potential increases in reported cases within 1–2 weeks. Although not directly comparable, other published studies have reported evidence that wastewater provides up to several weeks early warning of case trends or outbreaks of norovirus [8,18,31], SARS-CoV-2 [5,32], and hepatitis [18], overall confirming the usefulness of wastewater as a timely surveillance foundation.

For wastewater surveillance to act as a reliable public health tool, there is a specific need for it to accurately capture episodes of unusual activity—i.e., clusters or outbreaks—in addition to seasonal trends. We evaluated the ‘sensitivity’ of our surveillance program using temporal comparisons between peaks in wastewater concentrations (a pre-determined threshold being exceeded) and reported outbreaks of *Salmonella*, *Campylobacter,* and norovirus as well as gastrointestinal outbreaks of unknown etiology. Overall, we found that all weeks with a *Salmonella* wastewater alert correlated in time with either a salmonellosis outbreak with confirmed cases in Oklahoma, a reported multistate *Salmonella* outbreak without confirmed cases in Oklahoma, or a reported gastrointestinal outbreak of unknown etiology. Conversely, there was one confirmed *Campylobacter* outbreak in Oklahoma, but the timing of this did not correspond to any signals in wastewater. However, based on the wastewater concentrations alone, our results suggested as many as 10 weeks of *Campylobacter* outbreaks of which two correlated in time with gastrointestinal outbreaks of unknown etiology. These results highlight a possibility that some *Campylobacter* infections in Oklahoma remain undetected through traditional surveillance. Even though no norovirus outbreaks were officially reported in Oklahoma during the study period, concentrations in wastewater indicated a possible seven weeks of outbreak alerts. Of these, 71% occurred at the same time as a confirmed gastrointestinal outbreak of unknown etiology. Interestingly, the wastewater surveillance suggested that two US multistate gastrointestinal outbreaks (one *Salmonella* and one norovirus), which had no officially reported cases in Oklahoma, may also have impacted the state without cases being detected through the traditional surveillance methods.

Our study presents relevant insights into how wastewater monitoring can contribute to the surveillance of gastrointestinal infections in humans. However, the results need to be interpreted with consideration to potential limitations. Firstly, the assays used did not distinguish between individual species or serotypes of *Campylobacter* and non-typhoidal *Salmonella*. Depending on the accuracy of the assays to detect all species and serotypes causing human infection, our surveillance may have missed trends or clusters in specific species or serotypes which may explain the lack of correlation in time with some of the reported outbreaks. The data presented in this paper are statewide averages, with the number of wastewater sampling locations increasing from only 10 to 32 over the surveillance period. The geographical aggregation of data increases the risk of not detecting local increases in wastewater unless the increase was strong enough to significantly impact the statewide average. This is likely one of the factors explaining why several weeks of reported outbreaks were not visible in wastewater concentrations. Though the assays used in this program have been validated for the detection of *Salmonella*, *Campylobacter,* and norovirus in samples of non-human origin including wastewater [22,23,24], there is a risk that they are also detecting other pathogens/biomarkers because of cross-reactivity. Considering that we found a correlation between seasonal patterns in human cases and pathogen wastewater concentrations as well as the timing of wastewater alerts and reported outbreaks, we consider this limitation to not have significantly impacted the results.

As an important limitation, there is also the likelihood that our wastewater samples contained feces and urine from animals carrying *Campylobacter* or *Salmonella* spp. This could result in increasing or decreasing concentration trends which were unrelated to human cases. While the likelihood of animal waste entering the human wastewater system is generally low, it can happen in areas where livestock are concentrated and agricultural facilities are connected to municipal sewer systems. Based on the observed correlation between human cases and our wastewater concentrations, we believe that any confounding from animal feces and urine was low; however, it cannot be eliminated as a limitation.

Surveillance of human cases, and outbreaks, of gastrointestinal pathogens is universally delayed, meaning that cases are not diagnosed and reported for several weeks following actual infection date and symptom onset. In relation to wastewater monitoring, this significantly shifts the timing and can result in a lack of temporal correlation. We found that increases in wastewater precede increases in *Salmonella* and *Campylobacter* cases by one and two weeks, respectively, but this pattern is likely skewed by case notification delays. In addition to this, the relationship between reported cases and wastewater concentrations is also clouded by the fact that only a small percentage of infections are actually diagnosed and reported [1,2,33]. Therefore, peaks in wastewater concentrations may never reflect numbers of notified cases. This can explain our observation of several ‘outbreaks’ indicated in wastewater but not apparent in case reports. We also had a total of 18 weeks’ scheduled breaks in the surveillance which could have impacted the time match between reported outbreaks and those seen in wastewater. However, as these surveillance breaks were scheduled for the ‘off peak’ season (summer for norovirus and winter for *Salmonella* and *Campylobacter*), we consider them to have had minimal impact on the results. Finally, on any given day, the concentration of pathogens in wastewater will depend on the flow rate in sewers (a measure of population size combined with wastewater use). Because our results represent statewide averages, we did not adjust for potential daily flow fluctuations. However, it is important to note that regional or local measurements in pathogen concentrations are significantly influenced by wastewater flow differences [34,35].

Wastewater surveillance has proven to be a highly effective and useful monitoring tool during the COVID-19 pandemic, but evidence of its use for routine surveillance of other infectious diseases is limited. From our 18-month monitoring of gastrointestinal pathogens in wastewater in Oklahoma, we conclude that this alternative surveillance method has strong potential for detecting seasonal trends and case clusters or outbreaks. Additionally, wastewater appears to offer several weeks early warning of potential increases in human infections, compared to traditional disease surveillance. Lastly, our results suggest that several outbreaks of especially *Campylobacter* may go undetected by the current passive surveillance system in Oklahoma, potentially creating an unnecessarily high burden of this pathogen because those outbreaks are never reported and investigated.

## Figures and Tables

**Figure 1 microorganisms-11-02193-f001:**
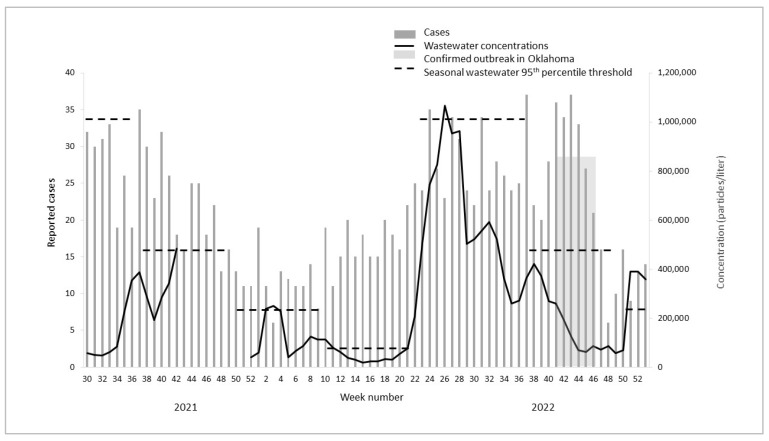
Reported cases, wastewater concentrations (three-week moving averages), and outbreak indicators for *Campylobacter* in Oklahoma, 2021–2022.

**Figure 2 microorganisms-11-02193-f002:**
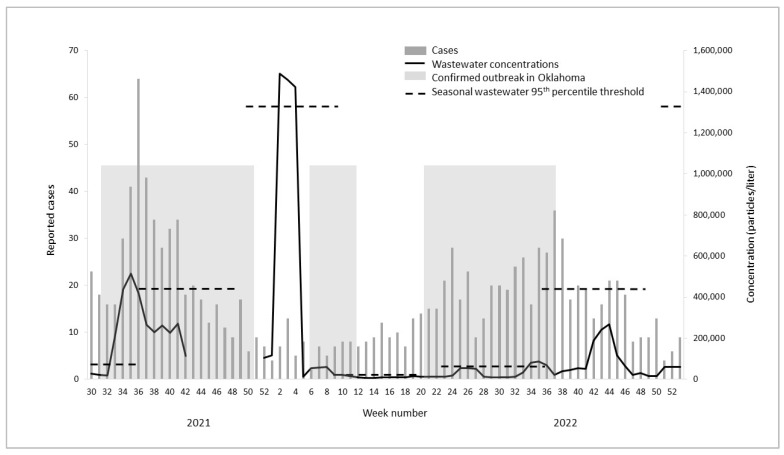
Reported cases, wastewater concentrations (three-week moving averages), and outbreak indicators for *Salmonella* in Oklahoma, 2021–2022.

**Figure 3 microorganisms-11-02193-f003:**
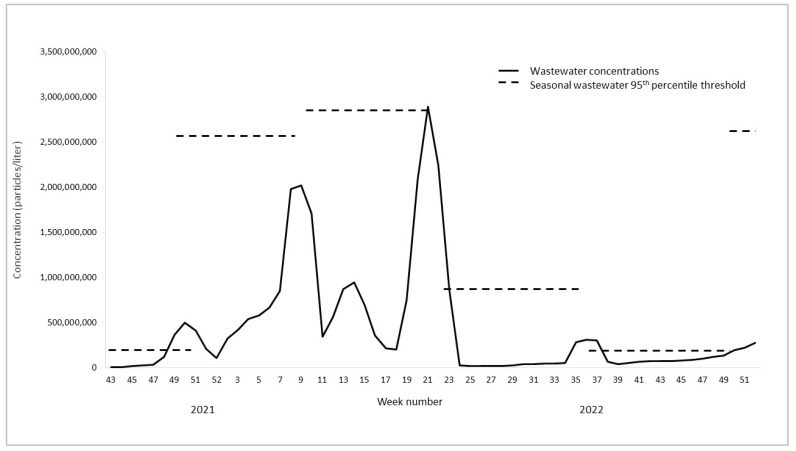
Wastewater concentrations (three-week moving averages) and outbreak indicators for norovirus in Oklahoma, 2021–2022.

**Table 1 microorganisms-11-02193-t001:** Reported cases and wastewater concentrations of *Campylobacter*, *Salmonella,* and norovirus in the state of Oklahoma, July 2021–December 2022.

Measure	*Campylobacter*	*Salmonella*	Norovirus
Total cases reported	1459	1157	n.a
Average weekly cases reported	22	17	n.a
Average weekly wastewater concentration *	180,000	127,000	390 million
Peak cases reported (date)	37	64	n.a
	(10 September 2022)	(4 September 2021)	
Peak wastewater concentration * (date)	1.7 million	4.2 million	4.5 billion
	(9 July 2022)	(22 January 2022)	(27 February 2022)

* Bacterial or viral particles per liter of wastewater. n.a. not applicable.

**Table 2 microorganisms-11-02193-t002:** Temporal association between outbreaks indicated in wastewater and reported outbreaks *, Oklahoma 2021–2022.

Pathogen	Number of Weeks with Outbreaks Reported	Number of Weeks Where Wastewater Concentrations Exceeded 95th Seasonal Threshold	Number of Weeks Where Exceeded Wastewater Thresholds Corresponded with Timing of Reported Outbreaks *	Number of Weeks Where Exceeded Wastewater Thresholds Corresponded with Timing of Outbreaks with Unknown Etiology * (N = 22)
*Campylobacter*	6	10	0	2
*Salmonella*	47	11	9	1
Norovirus	n.a	7	n.a	5
Total (%)	53	28	9 (32)	9 (41)

* Indicates whether the outbreak occurred no later than two weeks following the wastewater threshold being exceeded. n.a. not applicable.

**Table 3 microorganisms-11-02193-t003:** Temporal association between multistate outbreaks without reported cases in Oklahoma and Oklahoma wastewater surveillance thresholds, 2021–2022.

Pathogen	Multistate Outbreak, Week Number/Year Reported *	Oklahoma Wastewater Outbreak Threshold Exceeded, Week Number/Year
*Campylobacter*	None reported	n.a
*Salmonella*	*Salmonella* Senftenberg, 12–21/2022	19–20/2022
Norovirus	14–27/2022	21–23/2022

* Indicates whether the reported outbreak occurred no later than two weeks following the wastewater threshold being exceeded. n.a. not applicable.

## Data Availability

The data presented in this study are available upon request from the corresponding author and following permission from the funders. The data are not publicly available due to privacy restrictions.

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
