# Peer review of "Using Wastewater Surveillance to Monitor Gastrointestinal Pathogen Infections in the State of Oklahoma"

_microorganisms, 2023, doi:10.3390/microorganisms11092193_

Round 1

Reviewer 1 Report

The manuscript provides a comprehensive exploration of wastewater surveillance as a tool to monitor gastrointestinal pathogens in Oklahoma. The rationale for the study is articulated clearly, and the inclusion of anticipated pathogens broadens the study's scope effectively. The methodology, including the experiments and the comparative analysis between wastewater testing and actual case reports, is explained coherently.

I have a few minor suggestions mainly centered on the abstract and discussion sections. Aside from these, the manuscript is well-crafted and appears ready for publication.

Abstract:

To enhance its impact, consider including more specific findings in the results section, such as exact figures and p-values, which would provide a more concise summary of the study.

Discussion:

Line 246: Numerous studies have utilized wastewater for surveillance of various pathogens. To underscore the unique contribution of this study, please refine the sentence to clearly specify the pathogens being addressed. Furthermore, avoid reiterating the novel nature of this research multiple times in the discussion to minimize redundancy.

Line 261: I recommend rephrasing this line for brevity and clarity.

Line 275: In order for wastewater surveillance. -> For wastewater surveillance

Author Response

Response to Reviewer 1 comments

Thank you very much for your positive and encouraging comments. We appreciate your input. Please see below for specific responses to your individual comments.

Comment: Abstract. To enhance its impact, consider including more specific findings in the results section, such as exact figures and p-values, which would provide a more concise summary of the study.
Reply: Thank you, we have included numbers and p-values and revised the abstract to better summarise the results.

Comment: Line 246: Numerous studies have utilized wastewater for surveillance of various pathogens. To underscore the unique contribution of this study, please refine the sentence to clearly specify the pathogens being addressed. Furthermore, avoid reiterating the novel nature of this research multiple times in the discussion to minimize redundancy.

Reply: We agree with your observation and have re-worded this paragraph. We have also tried to minimize redundancy by deleting references to the novel nature of the research.

Comment: Line 261: I recommend rephrasing this line for brevity and clarity.

Reply: We have deleted this sentence.

Comment: Line 275: In order for wastewater surveillance. -> For wastewater surveillance

Reply: Thank you for the suggestion. We have amended the sentence.

Reviewer 2 Report

Comments to the Author

In general, this article has met all criteria required by the journal. Methodology, experimental design, explanation, and discussion of the results are presented clearly enough and not confuse the reader. The results can give a positive value in using wastewater surveillance to monitor gastrointestinal pathogen infections. However, there are some issues that need further elaboration.

1. What is the innovation of this manuscript?

2. Some other monitoring methods can be quoted in the introduction to contrast with wastewater monitoring and explain the advantages of wastewater monitoring.

3. What are the criteria for selecting a sewage treatment plant? (Line100-Line103)

4. Without access to data, how can we ensure the continuity of monitoring? (Line170-Line174)

5. The description of Figure 1, Figure 2, Figure 3 is not enough to fully reflect the picture information and needs further elaboration

6. When there is an outbreak of gastrointestinal pathogens in monitoring wastewater samples, there may already be cases upstream or in living areas. How to ensure the timeliness of wastewater monitoring?

7. It is suggested that the monitoring of physical and chemical properties of water samples should be added in the future monitoring to reflect the wastewater information more comprehensively.

8. What is the cause of “there were a number of wastewater outbreak alerts which did not match in time with reported gastrointestinal outbreaks in either Oklahoma or multi-state occurrences”? (Line240-Line242)

9. There are some grammatical errors, please check whole the manuscript for them.

Thanks

Author Response

Response to Reviewer 2 comments

Thank you for your comments and suggestions. Please see below for specific responses to your individual comments.

Comment 1. What is the innovation of this manuscript?

Reply: Thank you for the question. In the Introduction (lines 78-87), we highlight the lack of evidence for routine, long-term monitoring of gastrointestinal pathogens in wastewater. We further revisit the innovation of the study in the first lines of the Discussion (lines 246-254). We feel that the innovative contributions are adequately described and would prefer not to highlight this even more, at the risk of repeating ourselves and providing redundant information.

Comment 2. Some other monitoring methods can be quoted in the introduction to contrast with wastewater monitoring and explain the advantages of wastewater monitoring.

Reply: The first part of the Introduction (Lines 34-60) discusses the limitations of traditional passive surveillance (which is how all gastrointestinal pathogens are currently monitored) in comparison to wastewater. This is also reiterated in the last paragraph of the discussion. We are not quite sure which other surveillance methods you are referring to, as passive surveillance (symptoms-testing-reporting) is the classic and almost exclusively used one to which wastewater monitoring ideally should be compared.

Comment 3. What are the criteria for selecting a sewage treatment plant? (Line100-Line103)

Reply: Thank you for your question. We have specified these criteria in the paragraph.

Comment 4. Without access to data, how can we ensure the continuity of monitoring? (Line170-Line174)

Reply: Thank you for highlighting the concern with the scheduled breaks. These were unavoidable. We have included a mention of these breaks as a possible limitation in the Discussion (lines 344-348). As a note, we had scheduled these breaks during the ‘off peak’ season for each pathogen to minimize the chance of missing an outbreak.  

Comment 5. The description of Figure 1, Figure 2, Figure 3 is not enough to fully reflect the picture information and needs further elaboration.

Reply: We have added further explanation in the text to accompany the figures.

Comment 6. When there is an outbreak of gastrointestinal pathogens in monitoring wastewater samples, there may already be cases upstream or in living areas. How to ensure the timeliness of wastewater monitoring?

Reply: As mentioned in the Introduction and Discussion, wastewater surveillance has significant timing advantages compared to classic passive surveillance. Regardless of the location of cases, the wastewater treatment plant receives waste from all connected households in the catchment area as soon as the water has traveled through the system. Depending on the flow, this happens within hours. At the wastewater plant, our samples are collected as influent. That means we capture all excreted viral or bacterial particles on the same day as they are excreted by the infected person. Any increase in the number of particles, i.e. an outbreak signal, will be picked up on that day – from all persons living within the catchment area. Traditional surveillance usually picks up cases 2-3 weeks after they have been infected, because it relies on people seeking medical attention, being tested, diagnosed and their infection reported to the department of health.

Comment 7. It is suggested that the monitoring of physical and chemical properties of water samples should be added in the future monitoring to reflect the wastewater information more comprehensively.

Reply: Thank you for this suggestion. Our laboratory is equipped with expertise and facilities to test for pathogens but not general water properties. We are in the process of arranging for the wastewater treatment plants themselves to provide us with any data on water chemistry that they possess. We have not elaborated on this in the current manuscript because it is outside the scope of the study.

Comment 8. What is the cause of “there were a number of wastewater outbreak alerts which did not match in time with reported gastrointestinal outbreaks in either Oklahoma or multi-state occurrences”? (Line240-Line242)

Reply: We discuss this in Lines 296-307 and 337-346. Because only a small fraction of actual infections are reported (people do not always seek medical attention for gastrointestinal upsets), it is not surprising that wastewater indicates more outbreaks than are reported. Wastewater surveillance captures all people who are infected, regardless of whether they have symptoms or seek medical attention.

Comment 9. There are some grammatical errors, please check whole the manuscript for them.

Reply: Thank you for highlighting this. We have checked the whole manuscript and no grammatical errors were flagged. If you refer to the American English spelling of words, we have not been asked to write the manuscript in British English, however are happy to do so if required.

Round 2

Reviewer 2 Report

OK